# Factors affecting men's involvement in maternity waiting home utilization in North Achefer district, Northwest Ethiopia: A cross-sectional study

**Getachew Asmare[1], Dabere Nigatu👤[2]\*, Yamrot Debela[3]**

**1** Department of Reproductive Health and Nutrition, School of Public Health, College of Medicine and Health Sciences, Wolaita Sodo University, Wolaita Sodo, Ethiopia, **2** Department of Reproductive Health and Population Studies, School of Public Health, College of Medicine and Health Sciences, Bahir Dar University, Bahir Dar, Ethiopia, **3** Department of Health Promotion & Behavioral Science, School of Public Health, College of Medicine and Health Sciences, Bahir Dar University, Bahir Dar, Ethiopia

\* daberen@yahoo.com

## Abstract

### Background

Maternity waiting home (MWH) is a direct strategy to improve newborn and maternal survival. The utilization of MWH, however, remains very low in Ethiopia. Men involvement in maternal health programs is a key strategy to increase utilization of maternal health services, including MWH. This study defines men involvement in-terms of men's participation in deciding to admit their spouse to an MWH, accompanying their spouse to an MWH, providing financial support, availing food at an MWH, and taking care of the home or children. Thus, the current study aims to identify factors affecting men's involvement in MWH utilization.

### Methods

A community-based cross-sectional study was conducted from October 1st to December 30th, 2018. Four hundred three men were involved in the study. Data were analyzed by the statistical package for social science (SPSS) version 23. Independent predictors were identified by a multivariable logistic regression model. Adjusted odds ratios (AORs) with 95% confidence intervals (CIs) were reported.

### Results

Men's involvement in MWH was 55.6% (50.71, 60.45). Age (AOR = 0.86, 95% CI = 0.82–0.94), knowledge about MWH (AOR = 4.74, 95% CI = 2.65–8.49), decision-making power (AOR = 4.00, 95% CI = 1.38–11.57), and receiving counseling about MWH during spousal antenatal care visits (AOR = 9.15, 95% CI = 3.34–25.03) had statistically significant associations with men's involvement in MWH utilization.

**Data Availability Statement:** All relevant data are within the manuscript and its Supporting information files.

**Funding:** The author(s) received no specific funding for this work.

**Competing interests:** The authors have declared that no competing interests exist.

**Abbreviations:** ANC, Antenatal Care; AOR, adjusted odds ratio; CI, confidence intervals; COR, crude odds ratio; MWH, Maternity Waiting Home; SPSS, Statistical Package for Social Science.

## Conclusions

Nearly, half of the male partners were involved in MWH utilization. Men's age, MWH knowledge, decision-making power, and receiving counseling were factors affecting their involvement in MWH utilization. Interventions targeting to improve male involvement in MWH utilization should focus on building men's knowledge about MWH, increasing male involvement in ANC with an appropriate level of counseling about MWH, and changing patriarchal thinking in society with appropriate behavioral interventions such as community-based health education.

## Introduction

Global experiences show that more than 80% of maternal deaths could have been prevented by appropriate and timely interventions performed by skilled professionals in a conducive environment [1, 2]. In 2016, only 26% of women in Ethiopia gave birth at health facilities. This rate is among the lowest in the world. The reasons for non-use of skilled delivery service include notions that facility delivery is not necessary or customary, physical distance to the facility, and lack of transportation [3]. It has been many years since maternity waiting homes (MWHs) have been considered as a direct strategy for increasing health facility delivery and improving maternal and newborn survival [4, 5]. MWHs are residential facilities located near a hospital or a health center that allow pregnant women to wait for the onset of labor. Once labor starts, women move to the health facility so that they can be assisted by a skilled birth attendant [6]. Pregnant women from remote areas, women with a gestational age greater than 37 weeks, women with previous pregnancy/delivery problems (preterm labor, stillbirth, cervical tear), or women with other known risks are eligible for accommodation at MWHs [7, 8]. In 2016, a survey conducted in four regions of Ethiopia reported that 70% of health centers had MWHs [9] and nationally, about half of the facilities had MWHs [10].

Although MWHs commenced operations in the late 1980s in Ethiopia [6], service uptake remains low because of sociodemographic, economic, cultural, and gender- and facility-related constraints [7, 9]. Moreover, in most developing countries, including Ethiopia, most communities assign a low position to women, which makes them dependent on either collective decision-making with their partner or completely dependent on their partner's decision on issues that affect their health [11]. To overcome such problems, nearly two decades ago, the concept of male involvement in maternal health has been promoted as an essential element of the World Health Organization's initiative for making pregnancy and childbirth safer [12]. The rationale for seeking the involvement of men includes a view of men as gatekeepers and decision-makers for prompt access to health services, as responsible partners of women, as an important member of the community, and as their preference to be involved as fathers/partners [13]. For example, 55% of women in Ethiopia to 95% of women in Kenya need their husband's permission to use MWHs [14, 15], while approximately one-third (33%) of mothers in Ethiopia experienced refusal of admission by their husbands [9]. In general, the need for an increased engagement of men in reproductive, maternal, and child health is consistent with several global instruments that promote human rights and gender equity such as the 1994 International Conference on Population and Development program of action and the 1979 Convention on the Elimination of All Forms of Discrimination Against Women [16].

The term "male involvement" varies according to authors [17]. Male involvement in the context of maternal and child health refers to men's active involvement in the care of their partners and children [16, 18], or it is a broad concept that refers to the various ways in which men relate to reproductive health problems and programs, reproductive rights, and reproductive behavior [19]. Thus, the current study applied an inclusive definition for male involvement in MWHs [17, 20–23], which includes male participation in the decision to admit their spouse to an MWH, accompanying their spouse to an MWH, providing financial support, availing food at an MWH, and taking care of the home and/or the remaining children at home.

Evidence shows improvements in health outcomes where men are actively involved. A systematic review revealed that male involvement is associated with improved maternal health outcomes in developing countries [24]. In African countries, including Ethiopia, male involvement in antenatal care (ANC) and delivery is associated with increased spousal use of skilled birth attendant and postnatal care (PNC) [20, 22, 24–26]. There are also studies addressing factors influencing male involvement in maternal health services such as ANC, delivery, PNC and family planning [17, 18, 20, 21, 23, 27], while there is a lack of evidence about male involvement in MWH utilization. Hence, studying men's involvement in MWH utilization has paramount importance for policy-makers, programmers, and healthcare planners in designing evidence-based interventions. Therefore, this study aims to determine the extent of male involvement in MWH utilization and identify the factors that affect their involvement in Northwest Ethiopia.

## Methods

### Study design and settings

A community-based cross-sectional study was conducted in the North Achefer district from October 1st, 2018 to December 30th, 2018. The district is located in the West Gojjam Zone, Amhara regional state, Ethiopia. It has a total of 27 kebeles ("kebele" is the lowest administrative unit in Ethiopia). Regarding health infrastructure, it has one primary hospital, seven health centers, five private clinics and twenty-seven health posts. During the time of data collection, each of the health centers in the district had MWHs, but only five of the health centers had functional MWHs [28].

### Sample size and sampling procedure

Initially, we proposed to include 442 male partners in the study, but 403 male partners were involved at the end. A single population proportion formula was used to determine the sample size with the assumption of a 95% confidence level, 50% expected proportion of men involved, and 5% margin of error. The formula:

$$\mathbf{n} = \frac{\left(\mathbf{Z}_{\frac{\alpha}{2}}\right)^{2}\mathbf{p}(1-\mathbf{p})}{\mathbf{d}^{2}} = \frac{(1.96)^{2} \times 0.5(0.5)}{(0.05)^{2}} = 384$$

Where; n is the sample size, $Z_{\alpha/2}$ is critical value for normal distribution at 95% confidence level, p is the expected proportion, and d is the margin of error. We targeted to involve 442 men with consideration of 15% for non-responses. In the North Achefer district, only five health centers had functional MWHs. First, the principal investigator identified 662 mothers who had used MWH in the last one year from the maternity-waiting-home-users registration book. Then, residential profiles (kebeles and gotts "subdivision below kebele in Ethiopia") of mothers were identified from MWH registration books. Finally, mothers were selected by a

table of random numbers. Mothers from nearby districts, divorced and widows were excluded. Men who were living with their spouses were considered in the study.

## Study variables and measurements

Male partners were interviewed using a structured Amharic version questionnaire. The questionnaire was developed by reviewing different related literature [17, 20–23]. It was pretested on 5% of the sample size. The pretest was done in the nearby district, designated as South Achefer district. Then, the questionnaire was amended for wording, sequencing and content as the pretest output suggested. We have attached both the Amharic and English versions of the questionnaire as supporting information (S1 and S2 Files). Trained data collectors and supervisors were involved in the data collection process. The interviews were conducted in the respondents' residential houses. If the selected respondent was not available at the time of the first home visit, two re-visits were made.

The questionnaire comprised sociodemographic variables (age, educational status, wealth index, occupation and number of children); participant's spousal obstetric history (previous stillbirth, previous health facility delivery, length of stay at an MWH, ANC follow-up, history of spousal obstetric complication); health facility-related variables (basic social services, presence of ambulance, and daily follow-up at MWHs); male partner's gender thinking (number of wives, decision-making power); and male partner's knowledge and attitude towards MWH. The wealth index was created using principal component analysis. First, Pearson's correlation coefficients were determined for each item. An exploratory factor analysis was conducted to obtain the latent variables of the covariance structure. Then, the items were reduced to twelve factors based on the factor loadings, followed by re-analysis of the remaining factors. After that, the factor loadings and dispersal rate of all the factors were determined. Finally, the summative scores were divided into five equal groups (very poor, poor, middle, rich and very rich).

Five knowledge items were used to assess men's knowledge about MWH. All correct responses on five items were added to produce a composite index. We used eight items with a five-point Likert scale to assess men's attitudes towards MWH. The sum score was generated by adding individual scores on each item. Those men who scored above the median were considered to have a positive attitude towards MWH utilization otherwise taken as having a negative attitude [29–31].

The outcome variable for this study was men's involvement in MWH utilization. Six items were used to measure men's involvement in MWH utilization. The items used include male partner participation in deciding to rest their spouse to an MWH, accompanying their spouse to an MWH, providing financial support while their spouse stay at an MWH, availing food while their spouse stay at an MWH and taking care of the home and/or the remaining children while their spouse stay at an MWH. Each item has yes or no response options and coded 1 yes or 0 no. We added each item score to generate a composite index for male involvement in MWH utilization. Those men who scored less than three were considered as poor male involvement while those who scored greater than or equal to three were considered good male involvement.

## Data analysis

Data were checked, coded and entered into Epi-data version 3.1 and exported to SPSS version 23 for analysis. The reliability of items used to measure men's involvement in MWH utilization and knowledge and attitude towards MWH were checked by Cronbach's alpha value. The Cronbach's alpha value of the six items used to assess men's involvement in MWH utilization was 0.73, which is in acceptable range. Binary logistic regression analysis was used to

determine the association between explanatory variables and men's involvement in MWH utilization. Those candidate variables that were significant ($p<0.25$) in the bivariable analysis were entered into the multivariable logistic regression analysis. Finally, adjusted odds rations (AORs) with 95% confidence intervals (CIs) were used to identify independent predictors of men's involvement in MWH utilization.

This research paper is prepared following the "Strengthening the Reporting of Observational Studies in Epidemiology (STROBE)" checklist for cross-sectional study reporting guidelines [32] (S1 Table).

## Ethical considerations

Ethical clearance letter was obtained from Institutional Review Board of Bahir Dar University College of Medicine and Health Sciences. The permission letter was obtained from the North Achefer district administrative. Moreover, all the study participants were informed about the purpose and benefit of the study along with their right to refuse. The data collectors read the information sheet and consent form to each study participant until they comprehend the contents. Then, the participants were supposed to show their agreement or disagreement verbally instead of hand signed consent approval. Finally, the data collectors are supposed to circle on the appropriate response of the participant to proceed to the next step. The study participants were reassured to attain confidentiality. We maintained anonymity and confidentiality of information throughout the study process.

## Results

### Sociodemographic and economic characteristics of the study participants

Four hundred three male partners were involved in the study, resulting in a response rate of 91.2%. Thirty-seven per cent of males were between the age groups of 40–49. The majority, 96.8%, of males were orthodox Christian followers. Ninety-two per cent were Amhara by ethnicity. Eighty-one per cent of males were farmers. Approximately 49% of males were unable to read and write. Nearly 22% of males were rich and 20.8% were poor (Table 1).

### Obstetric histories of wives

Approximately, 77% of wives gave birth before the current child. About 66.8% of wives had a previous history of health facility delivery and 12.8% had a previous history of obstetric complications. The commonest obstetric complications were hemorrhage (37.5%) and prolonged labor (37.5%). Eighty-six per cent of wives stayed less than fifteen days at MWH for the current child. Eighty-two per cent of wives had ANC follow-up for the current child, of whom 73% of men accompanied their spouse during ANC visit and 65.9% of men received counseling about MWH (Table 2).

### Men's knowledge and attitude towards MWH and gender thinking

Male partners were asked about gender thinking that likely influences their involvement in resting pregnant women in MWH: 17.1% think that childbirth is woman's affair that does not require the participation of men, 16.1% think that childbirth is a natural phenomenon that should not require much attention from men, and 28.8% think that accompanying wife to an MWH is a woman's responsibility. Forty-four per cent of men were sole decision-maker in any family affairs. Almost all, 99% of men had monogamous marriage (Table 3).

**Table 1. Sociodemographic characteristics of study participants in North Achefer district, Northwest Ethiopia, 2018.**

| Variables | Frequency | Percentage |
|---|---|---|
| **Age category (n = 403)** | | |
| 18–29 | 77 | 19.1 |
| 30–39 | 91 | 22.6 |
| 40–49 | 149 | 37.0 |
| > = 50 | 86 | 21.3 |
| **Educational status (n = 403)** | | |
| Unable to read and write | 196 | 48.6 |
| Read and write only | 82 | 20.3 |
| Primary education | 65 | 16.1 |
| Secondary education | 12 | 3.1 |
| Higher education | 48 | 11.9 |
| **Occupation (n = 403)** | | |
| Farmer | 326 | 80.9 |
| Merchant | 27 | 6.7 |
| Government employee | 50 | 12.4 |
| **Number of living children (n = 403)** | | |
| 1 | 94 | 23.3 |
| 2–4 | 138 | 34.3 |
| > = 5 | 171 | 42.4 |
| **Wealth index quintile (n = 403)** | | |
| Very rich | 80 | 19.9 |
| Rich | 88 | 21.8 |
| Middle | 74 | 18.4 |
| Poor | 84 | 20.8 |
| Very poor | 77 | 19.1 |

The Cronbach's alpha value of the knowledge questions was 0.755. The Cronbach's alpha value of the attitude questions was 0.127. As indicated in Table 3, 36.7% of men had a positive attitude.

## Men's involvement in maternity waiting home

The Cronbach's alpha value of the items used to measure men's involvement in MWH was 0.73. The mean and median of men involvement scores were 3.26 and 4 respectively.

Overall, 55.6% of male partners had good involvement in MWH utilization (Fig 1). Findings from specific indicators of male involvement show that 56.3% of men had decided to rest their spouses at an MWH, 54.1% accompanied their spouse to an MWH, 52.6% provided financial support while their spouses stayed at an MWH, 62.5% availed food while their spouses stayed at an MWH and 45.9% looked after the home and/or the remaining children while their spouses were at an MWH (Table 4).

## Factors influencing men's involvement in MWH utilization

A multivariable logistic regression model was fitted to identify predictors of men involvement in MWH utilization. In the bivariable logistic regression analysis, variables with p-values less than 0.25 were considered as candidate variables for the multivariable logistic regression model. Thus, age, occupation, educational status, number of live children, wealth index,

**Table 2. Study participant's spousal obstetric history in North Achefer district, 2018.**

| Variables | Frequency | Percentage |
|---|---|---|
| **Previous delivery history (n = 403)** | | |
| Yes | 312 | 77.4 |
| No | 91 | 22.6 |
| **Previous health facility delivery (n = 403)** | | |
| Yes | 209 | 66.8 |
| No | 104 | 33.2 |
| **Previous history of obstetric complication (n = 403)** | | |
| Yes | 40 | 12.8 |
| No | 273 | 87.2 |
| **Types of obstetric complications (n = 40)** | | |
| Preterm labour | 2 | 5.0 |
| Premature rapture of membrane | 8 | 20.0 |
| Hemorrhage | 15 | 37.5 |
| Prolonged labor | 15 | 37.5 |
| **Previous stillbirth history (n = 403)** | | |
| Yes | 11 | 3.5 |
| No | 302 | 96.5 |
| **Duration of stay at MWHs (n = 403)** | | |
| < 7 days | 247 | 61.3 |
| 7–13 days | 93 | 23.1 |
| > = 14 days | 63 | 15.6 |
| **Spousal ANC visit (n = 403)** | | |
| No visit | 72 | 17.9 |
| 1–3 visits | 104 | 25.8 |
| > = 4 visits | 227 | 56.3 |
| **Male partner accompaniment during ANC visit (n = 331)** | | |
| Yes | 255 | 73.0 |
| No | 76 | 23.0 |
| **Male partners got counseling about MWH during ANC visit (n = 331)** | | |
| Yes | 218 | 65.9 |
| No | 113 | 34.1 |

MWH maternity waiting home, ANC antenatal care.

distance from MWH, duration of stay at MWH, knowledge about MWH, attitude towards MWH, decision-making power, spousal ANC follow-up, previous spousal health facility delivery, previous spousal obstetric complication, and receiving counseling about MWH during spousal ANC follow-up were entered into the multivariable model. There were variables with wide confidence intervals in the model. This might be explained by sample size adequacy and presence of cells with small observation.

Based on findings from multivariable logistic regression analysis, a year increase in age was associated with a 14% decrease in the likelihood of men's involvement in MWH utilization (AOR = 0.86, 95% CI = 0.82–0.94). MWH knowledge of male partners was positively associated with their involvement in MWH utilization. A unit increase in MWH knowledge score was associated with 4.74 times increase in the likelihood of involvement in MWH utilization (AOR = 4.74, 95% CI = 2.65–8.49). Men who were a primary decision-maker in family affair were 4 times more likely to be involved in MWHs compared to those who have made shared

**Table 3. Men's gender thinking and attitude towards MWH in North Achefer district, 2018.**

| Variables | Frequency | Percent |
|---|---|---|
| **Child-birth is a woman's affair that does not require men participation (n = 403)** | | |
| Yes | 69 | 17.1 |
| No | 334 | 82.9 |
| **Child-birth is natural phenomenon that should not require much attention from men (n = 403)** | | |
| Yes | 65 | 16.1 |
| No | 338 | 83.9 |
| **Accompanying wife to MWH before delivery is a woman's responsibility (n = 403)** | | |
| Yes | 116 | 28.8 |
| No | 287 | 71.2 |
| **Who is the primary decision-maker in your family in any case that needs decision? (n = 403)** | | |
| Male alone | 179 | 44.4 |
| Wife alone | 50 | 12.4 |
| Spouses jointly | 174 | 43.2 |
| **Male partners having more than one wife (n = 403)** | | |
| Yes | 4 | 1.0 |
| No | 399 | 99.0 |
| **Attitude towards MWH (n = 403)** | | |
| Positive attitude | 148 | 36.7 |
| Negative attitude | 255 | 63.3 |

MWH maternity waiting home.

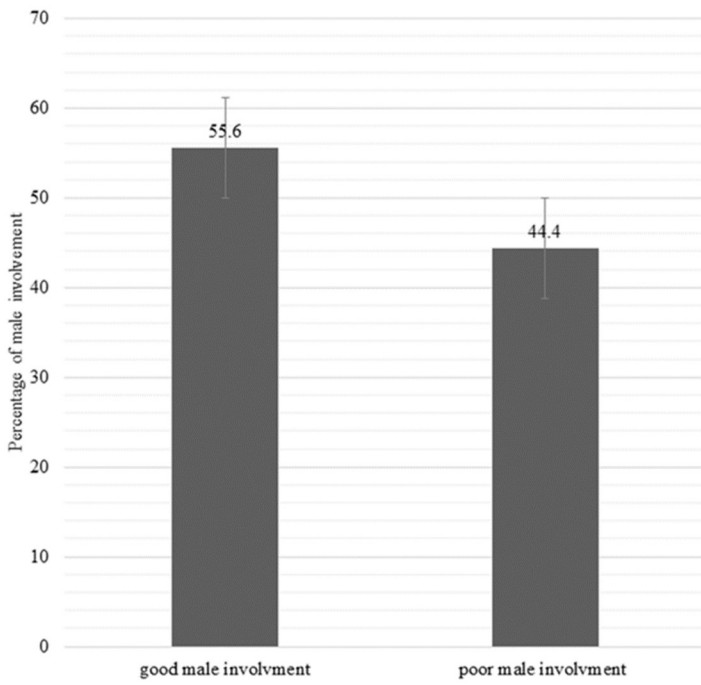

**Fig 1. Overall men's involvement in maternity waiting home utilization, North Achefer district, 2018.**

**Table 4. Distribution of men's involvement in MWH utilization in North Achefer district, 2018.**

| Variables | Frequency | Percent |
|---|---|---|
| **Decided to admit their spouse in MWH for current child (n = 403)** | | |
| Yes | 227 | 56.3 |
| No | 176 | 43.7 |
| **Accompanied their spouse to an MWH for current child (n = 403)** | | |
| Yes | 218 | 54.1 |
| No | 185 | 45.9 |
| **Provided financial support for their spouse while they went to/were at MWH for the current child (n = 403)** | | |
| Yes | 212 | 52.6 |
| No | 191 | 47.4 |
| **Availed food when their spouse and relatives were at MWHs (n = 403)** | | |
| Yes | 252 | 62.5 |
| No | 151 | 37.5 |
| **Looked after the home and/or children while their spouses were at MWHs for the current child (n = 403)** | | |
| Yes | 185 | 45.9 |
| No | 218 | 54.1 |
| **Arranged transport when their spouse went to MWHs (n = 403)** | | |
| Yes | 218 | 54.1 |
| No | 185 | 45.9 |

MWH maternity waiting home.

decision (AOR = 4.00, 95% CI = 1.38–11.57). Those male partners who have received counseling about MWH during spousal ANC follow-up were 9 times more likely to involve in MWH utilization compared to those who have not received counseling (AOR = 9.15, 95% CI = 3.34–25.03) (Table 5).

**Table 5. Factors affecting male partners' involvement in MWH utilization in North Achefer district, 2018.**

| Variables | Male involvement | | COR (95% CI) | AOR (95% CI) |
|---|---|---|---|---|
| | Good | Poor | | |
| **Age in year** | | | 0.86(0.84, 0.89) | 0.86(0.82,0.94)** |
| **MWH Knowledge score** | | | 8.57(5.29,13.87) | 4.74(2.65,8.49)** |
| **Decision maker in family affair** | | | | |
| Male alone | 31.8% | 12.7% | 2.18(1.41,3.40) | 4.00(1.38,11.57)* |
| Wife alone | 0.7% | 11.7% | 0.06(0.02,0.19) | 0.29(0.05,1.75) |
| Partners jointly | 23.1% | 20% | 1 | 1 |
| **Received counseling about MWH** | | | | |
| Yes | 55% | 10.9% | 19.78(11.07,35.36) | 9.15(3.34,25.03)** |
| No | 6.9% | 27.2% | 1 | 1 |

COR crude odds ratio, AOR adjusted odds ratio,

* indicates variables that are significant at p<0.05,

** indicates variables that are significant at p<0.001,

MWH maternity waiting home.

## Discussion

The study revealed that men involvement in MWH utilization was 55.6% with 95% CI (50.71–60.45). Our study also identified that age, knowledge towards MWH, decision-making autonomy and receiving counseling about MWH were factors significantly influencing men's involvement in MWH utilization.

This study revealed that a small proportion of men were involved in MWH. It is assumed low because once a pregnant woman admitted in an MWH, she is supposed to stay there until labor starts. The duration of stay at MWH may range from few days to many weeks. A study done in Ethiopia reported that on average, pregnant women stayed 14.8 days at the MWHs, and approximately 40% of pregnant women stayed for two or more weeks [9]. In the current study also about 16% of pregnant women stayed two or more weeks at MWH and on average they stayed more than a week. The longer the women stay at MWH the more they seek the support of their male partners. If this is not achieved, it could have a negative implication on future use of MWH.

This study identified that an increase in men age was associated with a decrease in men involvement in MWH utilization. This might be due to the fact that as men get older and older, they might develop patriarchal thinking and uncaring attitude for their wife and would be born child. This finding is consistent with a study done in Lemo woreda of Ethiopia [33].

The current study found that men's knowledge about MWH was associated with increased involvement in MWH utilization. Similarly, our study noted a positive association between receiving counseling about MWH during spousal ANC follow-up and male involvement in MWH utilization. This might be because having knowledge is a prerequisite for practice. If men have awareness about benefit packages of MWH through different outlets, including via health worker counseling, they could be encouraged to be involved in service uptake. This finding is consistent with other studies done to assess men's involvement in delivery services and in birth preparedness and complication readiness plan in Southern Ethiopia, Lemo district of Ethiopia, Ambo town of Ethiopia, Mekelle town of Ethiopia, Enderta district of Ethiopia, Kenya, India, Mali and Tanzania [14, 23, 34–41].

Men's sole decision-making in family affairs was positively associated with male involvement in MWH utilization. This finding implies the presence of male dominance in society. The current finding is in-line with a qualitative study done in Zambia [34]. This might be assumed that whenever men are the primary decision makers in a family, they will have the power to allow or refuse their spouses to utilize maternal health services. Men being primary drivers of decision, in turn, might have a cultural implication of male dominance attitude and gender stereotypic outlook in the society, which are a base for gender inequalities and gender-based violence.

The study has limitations. The lack of standardized indicators/tools to measure men's involvement in MWH utilization may be a limitation of the study. But we have developed the questionnaire through review of related literature and pretested the tool before actual study. In addition, this study is generalizable to male partner whose spouse have used MWH for the most recent birth. The findings can also be generalizable to other similar settings in Ethiopia and outside of Ethiopia. The study, however, cannot tell us the extent of male involvement for those women who have not used MWH.

## Conclusions

Nearly, half of male partners showed poor involvement in MWH utilization. Men's knowledge towards MWH, receiving counseling during spousal ANC visits, men's sole decision-making in family affairs, and being younger age were factors positively influencing men's involvement

in MWH utilization. Interventions targeting to improve male involvement in MWH utilization should focus on building men's knowledge about MWH, increasing male involvement in ANC with an appropriate level of counseling about MWH, and changing patriarchal thinking in society through appropriate behavioral interventions such as community-based health education intervention.

## Supporting information

**S1 Table. STROBE 2007 (v4) statement—Checklist of items that should be included in reports of cross-sectional studies.**
(PDF)

**S1 File. Amharic language version questionnaire.**
(PDF)

**S2 File. English language version questionnaire.**
(PDF)

## Acknowledgments

We are very grateful to Bahir Dar University for giving us ethical clearance. We are also indebted to thank the North Achefer district health office, kebele administrations and study participants for their cooperation during data collection.

## Author Contributions

**Conceptualization:** Getachew Asmare, Dabere Nigatu, Yamrot Debela.

**Data curation:** Getachew Asmare.

**Formal analysis:** Getachew Asmare.

**Investigation:** Getachew Asmare.

**Methodology:** Getachew Asmare, Dabere Nigatu, Yamrot Debela.

**Project administration:** Getachew Asmare.

**Resources:** Getachew Asmare.

**Supervision:** Dabere Nigatu, Yamrot Debela.

**Validation:** Getachew Asmare.

**Writing – original draft:** Getachew Asmare, Dabere Nigatu.

**Writing – review & editing:** Getachew Asmare, Dabere Nigatu, Yamrot Debela.

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
