## [Decision Letter · Decision Letter 0]

13 Jul 2021

PONE-D-20-07872

Factors affecting men’s involvement in maternity waiting home care in North Achefer district, Northwest Ethiopia: a cross-sectional study

PLOS ONE

Dear Dr. Nigatu,

Thank you for submitting your manuscript to PLOS ONE. After careful consideration, we feel that it has merit but does not fully meet PLOS ONE’s publication criteria as it currently stands. Therefore, we invite you to submit a revised version of the manuscript that addresses the points raised during the review process.

Please accept my apologies for the delay in communicating this decision to you.

The manuscript has been evaluated by three reviewers, and their comments are available below. Please note that as per our publication criteria, PLOS ONE requires that all experiments, statistics and other analyses are performed to a high technical standard, described in sufficient detail and adhere to appropriate reporting guidelines and community standards. Conclusions must be presented in an appropriate fashion and be supported by the data (Please see http://journals.plos.org/plosone/s/criteria-for-publication).

The reviewers have requested further details regarding the questionnaires used and statistics performed and have requested updates to a number of the conclusions within the manuscript. Please note that further context to Reviewer 1’s comments can be found in the attached document.

Please revise the manuscript to carefully address the concerns raised.

We look forward to receiving your revised manuscript.

Kind regards,

George Vousden

Division Editor

PLOS ONE

Journal Requirements:

A clean copy of the edited manuscript (uploaded as the new *manuscript* file).

4. Please upload a new copy of Figure 1 as the detail is not clear. Please follow the link for more information: https://blogs.plos.org/plos/2019/06/looking-good-tips-for-creating-your-plos-figures-graphics/" https://blogs.plos.org/plos/2019/06/looking-good-tips-for-creating-your-plos-figures-graphics/.

Reviewers' comments:

Reviewer's Responses to Questions

**Comments to the Author**

1. Is the manuscript technically sound, and do the data support the conclusions?

Reviewer #1: Yes

Reviewer #2: Yes

Reviewer #3: Partly

2. Has the statistical analysis been performed appropriately and rigorously? 

Reviewer #1: Yes

Reviewer #2: Yes

Reviewer #3: No

3. Have the authors made all data underlying the findings in their manuscript fully available?

Reviewer #1: No

Reviewer #2: Yes

Reviewer #3: Yes

4. Is the manuscript presented in an intelligible fashion and written in standard English?

Reviewer #1: Yes

Reviewer #2: Yes

Reviewer #3: No

5. Review Comments to the Author

Reviewer #1: The introduction miss the following

1. Description of male involvement as used in this study is missing

2. The state of male involvement in maternity waiting homes

3. What others have reported as far as male involvement is concern

4. What evidence are there in relationship between male involvement and use?

It is not clear what authors are referring to when they talk about knowledge. It created questions which call for clarity

1. knowledge about what?

2. How was it assessed

The same applies to attitudes

Reviewer #2: Reviewer Comments:

1. Is the manuscript technically sound, and do the data support the conclusions? The manuscript must describe a technically sound piece of scientific research with data that supports the conclusions. Experiments must have been conducted rigorously, with appropriate controls, replication, and sample sizes. The conclusions must be drawn appropriately based on the data presented.

I believe that the data collected and the analyses performed support the findings and the conclusions. A few comments are detailed below:

Study sample, page 5, line 100-101: “Males who were currently living with their spouse were considered in the study.” Were males who did not live with their spouse excluded? This is not clear. If males who did not live with their spouse were excluded, why? It would be interesting to include those males, assuming that decision-making may be different in those households or maybe males in those households were less involved?

Results, page 15, line 220: “…a one-year increase in men’s age was associated with a 12% decrease in the likelihood of men involvement in MWH utilization” – This finding is really interesting! As you have explained in the discussion, this may be due to the shift in attitude we are seeing in the newer generations.

Discussion, page 16, line 246-248: Please clarify this sentence. Do you mean that without support from the male partner, the woman will likely have a bad experience with a long MWH stay, therefore, she will likely not want to use the MWH nor recommend others to use MWHs? Maybe explain further on why the woman will have a bad experience without support from her male partner. My understanding is that female escorts usually accompany a pregnant woman during her MWH stay but that male partners provide financially as it is also indicated in the manuscript as part of the support.

Discussion, page 17, line 273: Please expand on the study’s limitations. I do not believe this is the only limitation of the study. I would suggest discussing the generalizability of the study. When discussing limitations, you can also comment on how those limitations may have been mitigated, if possible – for example, the lack of a standardized tool was mentioned as a limitation, but you also mentioned in the methods section that you pretested the questionnaire.

Conclusion, page 17, future interventions: Future interventions could also focus on increasing uptake of male involvement in ANC visits (currently at 73% which is pretty good already), particularly the 4th visit where they get counseling about MWHs, and increase MWH counseling at ANC visit by health facility staff (currently 65.9%). Male partners who received MWH counseling were 9 times more likely to be involved in MWH utilization compared to those who did not.

2. Has the statistical analysis been performed appropriately and rigorously?

The statistical analyses undertaken has been performed appropriately to my knowledge. Cronbach’s alpha is appropriately used for this study as the questionnaire included questions using the Likert scale to assess male partner’s involvement in MWH utilization. Additional comments are detailed below:

Methods, data analysis: What did you do with the questionnaires that were incomplete? Were they included or excluded?

Table 1, 2, 3, & 4: Please include sample size (n=) below frequency in the tables.

3. Have the authors made all data underlying the findings in their manuscript fully available? The data should be provided as part of the manuscript or its supporting information, or deposited into a public repository. If there are restrictions on publicly sharing data – e.g. participant privacy or use of data from a third party – those must be explained.

The authors indicated that all data are fully available without restriction and can be found within the manuscript and in the supporting information files.

4. Is the manuscript presented in an intelligible fashion and written in standard English? Any typographical or grammatical errors should be corrected at revision, if so please note any specific errors here.

Yes, this manuscript is presented in an intelligible fashion and written in standard English. Specific errors/corrections are noted below:

General, throughout manuscript: Maternity waiting home should be plural throughout the manuscript – maternity waiting homes (MWHs) – where appropriate. Since you gave the abbreviation MWH/MWHs, you can use it instead of writing out maternity waiting home or maternity waiting homes throughout the manuscript.

Page 3, line 45: Add “It has been” to the beginning to the sentence to make it complete. “It has been many years since maternity waiting homes (MWHs) were articulated as...”

Page 3, line 55: “Nationally” should be lower case.

Page 3, line 60: “depend on their partner’s decision...”

Page 4, line 66: “…men’s preference to be involved as fathers/partners.”

Page 4, line 81: “…data collection, each of the seven health centers in the district had a MWH, but only five...”

Page 4, line 83-85: “This reach paper is prepared following…” – This sentence seems to be out of place, maybe put it at the end of the methods section before ethical considerations?

Page 5, line 87: “Initially, we proposed…” change primarily to initially.

Page 5, line 97: What are gotts?

Page 5, line 99: “Maternity-home-user mothers from nearby districts, those who were divorced, and those who had husbands who were deceased were excluded.”

Page 5, line 107: “We have attached both the Amharic and English versions of the questionnaire as supporting information…”

Page 6, line 115: Remove “_” from “social_services”

Page 7, line 164: “Nearly 22% of males were rich…”

Page 9, line 174-176: Be consistent about which term you want to use for men/respondents/male partners. In the same sentence, three different terms are used. Remove apostrophe from “73% of men’s…” on line 175.

Page 11, line 182: “17.1% think that childbirth is a woman’s affair…”

Page 11, line 185: “…accompanying their wife to a MWH to stay before delivery is a woman’s responsibility.” Your variable “accompanying wife to MWH for delivery is a woman’s responsibility” suggests that women deliver at the MWH, but is that true? Women are supposed to be moved to the health facility when labor comes so they deliver in the health facility. MWHs are used only for them to stay before delivery.

Page 13, line 199-200: “56.3% of partners had decided to rest their spouse at a MWH…”

Page 13, line 201: “…provided financial support while their spouse stayed at a MWH…”

Table 4: Variable “accompanied their spouse for resting in MWH for current child”. Use past tense since your other variables are past tense.

Page 15, line 222: “MWH knowledge of male partners was positively associated…”

Page 15, line 221-222: “…associated with a 12% decrease…” AOR = 0.88 is written in text but in Table 5, AOR = 0.86

Page 15, line 225-226: “…associated with a 4.75 times increase…” AOR = 4.75 is written in text but in Table 5, AOR = 4.74

Page 16, line 242: “…assumed low because women are staying at a MWH until labor starts…” delete ‘s from “women’s”.

Page 17, line 271: “…whenever males are the primary decision makers in a family…” When it says “most decision makers” do you mean to say men are the primary decision makers in the household?

Reviewer #3: The manuscript titled, “Factors affecting men’s involvement in maternity waiting home care in North Achefer district, Northwest Ethiopia: a cross-sectional study” requires additional copy editing for English grammar and language. Double check the correct terminology for use versus utilization throughout.

Study variables and measurements

More details need to be provided for how the survey tool was developed. There are several questions in the supplementary file that are not addressed in the presentation of results. Variables are poorly defined. In section 2, wealth is extensively surveyed yet not described in the text. How were wealth quintiles defined? Were responses to questions in Sections 5,6,7 open ended or multiple choice? How were these survey questions derived? Was there any qualitative research guiding survey response choices? I’m not convinced that availing food when their spouse and relatives were at MWHs appropriately measures male partners’ involvement in MWHs. More details are needed about how Likert scales were determined and tested.

Results

Justify the use of so many variables in one multivariable model. In Table 2, consider stratifying duration of stay at MWH and how many ANC visits a woman and spouse attended. Provide rationale for how attitudes were labeled positive or negative. Several results had very wide confidence intervals and potential reasons for the wide intervals need to be offered.

Discussion

Be sure to discuss limitations of the study. Elaborate on the cultural implications of men being the main drivers of decision making. Finally, propose next steps and tell the reader how interventions can “change patriarchal thinking in society” as suggested in the final sentence.

6. PLOS authors have the option to publish the peer review history of their article (what does this mean?). If published, this will include your full peer review and any attached files.

Reviewer #1: **Yes: **Fabiola Moshi

Reviewer #2: No

Reviewer #3: No

---

## [Author Response · Author response to Decision Letter 0]

6 Oct 2021

Responses to reviewers

Response to reviewer #1

Comment #1: The introduction miss the following:

1. Description of male involvement as used in this study is missing

2. The state of male involvement in maternity waiting homes

3. What others have reported as far as male involvement is concern

4. What evidence are there in relationship between male involvement and use?

Response #1: Thank for the comment. We considered the comment in the revised manuscript. But still the authors did get direct literature done on male involvement in MWH instead we reviewed related literature.

Comment #2: It is not clear what authors are referring to when they talk about knowledge. It created questions which call for clarity

1. knowledge about what?

2. How was it assessed

The same applies to attitudes

Response #2: We have revised as per the comment. 

Response to reviewer #2

Comment #1: Study sample, page 5, line 100-101: “Males who were currently living with their spouse were considered in the study.” Were males who did not live with their spouse excluded? This is not clear. If males who did not live with their spouse were excluded, why? It would be interesting to include those males, assuming that decision-making may be different in those households or maybe males in those households were less involved?

Response #1: Thank you for this comment, the authors also believe that they might have different character in-terms of decision making, but the outcome of interest of the study is not decision making rather it is male involvement. In fact, the actual number of men who were excluded because of divorce were very few in number, only five males with formal divorce were excluded. Hence, it may not have significant impact on male involvement too.

Comment #2: Results, page 15, line 220: “…a one-year increase in men’s age was associated with a 12% decrease in the likelihood of men involvement in MWH utilization” – This finding is really interesting! As you have explained in the discussion, this may be due to the shift in attitude we are seeing in the newer generations.

Response #2: Thank for emphasizing this finding.

Comment #3: Discussion, page 16, line 246-248: Please clarify this sentence. Do you mean that without support from the male partner, the woman will likely have a bad experience with a long MWH stay, therefore, she will likely not want to use the MWH nor recommend others to use MWHs? Maybe explain further on why the woman will have a bad experience without support from her male partner. My understanding is that female escorts usually accompany a pregnant woman during her MWH stay but that male partners provide financially as it is also indicated in the manuscript as part of the support.

Response #3: Thank you for the comment, we have made revision for clarity of the referred paragraph. The authors interest here is that whenever pregnant women stay for long duration at MWH, they may expect or need to receive more support from their husband irrespective of other accompaniers presence with them. If they didn’t receive support from their husband as expected, this might have bad implication for future use of MWH. 

Comment #4: Discussion, page 17, line 273: Please expand on the study’s limitations. I do not believe this is the only limitation of the study. I would suggest discussing the generalizability of the study. When discussing limitations, you can also comment on how those limitations may have been mitigated, if possible – for example, the lack of a standardized tool was mentioned as a limitation, but you also mentioned in the methods section that you pretested the questionnaire.

Response #4: We have made changes accordingly.

Comment #5: Conclusion, page 17, future interventions: Future interventions could also focus on increasing uptake of male involvement in ANC visits (currently at 73% which is pretty good already), particularly the 4th visit where they get counseling about MWHs, and increase MWH counseling at ANC visit by health facility staff (currently 65.9%). Male partners who received MWH counseling were 9 times more likely to be involved in MWH utilization compared to those who did not.

Response #5: Thank you for your remainder to consider in the recommendation. We have considered in the revised version. 

Comment #6: Methods, data analysis: What did you do with the questionnaires that were incomplete? Were they included or excluded?

Response #6: The completeness was checked with field supervision. 

Comment #7: Table 1, 2, 3, & 4: Please include sample size (n=) below frequency in the tables.

Response #7: We have done so as per the comment.

Comment #8: General, throughout manuscript: Maternity waiting home should be plural throughout the manuscript – maternity waiting homes (MWHs) – where appropriate. Since you gave the abbreviation MWH/MWHs, you can use it instead of writing out maternity waiting home or maternity waiting homes throughout the manuscript.

Response #8: Thank you, revised accordingly.

Comment #9: Page 3, line 45: Add “It has been” to the beginning to the sentence to make it complete. “It has been many years since maternity waiting homes (MWHs) were articulated as...”

Response #9: Done a per the comment

Comment #10: Page 3, line 55: “Nationally” should be lower case.

Response #10: Thank you, done.

Comment #10: Page 3, line 60: “depend on their partner’s decision...”

Response #10: Thank you, done.

Comment #11: Page 4, line 66: “…men’s preference to be involved as fathers/partners.”

Response #11: Thank you, done.

Comment #12: Page 4, line 81: “…data collection, each of the seven health centers in the district had a MWH, but only five...”

Response #12: Thank you, done.

Comment #13: Page 4, line 83-85: “This reach paper is prepared following…” – This sentence seems to be out of place, maybe put it at the end of the methods section before ethical considerations?

Response #13: Thank you, done.

Comment #14: Page 5, line 87: “Initially, we proposed…” change primarily to initially.

Response #14: Thank you, done.

Comment #15: Page 5, line 97: What are gotts?

Response #15: Gotts are subdivision below kebele in Ethiopia. In Ethiopia, there are regions at the top----Zones----Districts----Kebeles----gotts

Comment #16: Page 5, line 99: “Maternity-home-user mothers from nearby districts, those who were divorced, and those who had husbands who were deceased were excluded.”

Response #16: Thank you, done.

Comment #17: Page 5, line 107: “We have attached both the Amharic and English versions of the questionnaire as supporting information…”

Response #17: Thank you, done.

Comment #18: Page 6, line 115: Remove “_” from “social_services”

Response #18: Thank you, done.

Comment #19: Page 7, line 164: “Nearly 22% of males were rich…”

Response #19: Thank you, done

Comment #20: Page 9, line 174-176: Be consistent about which term you want to use for men/respondents/male partners. In the same sentence, three different terms are used. Remove apostrophe from “73% of men’s…” on line 175.

Response #20: We have revised accordingly.

Comment #21: Page 11, line 182: “17.1% think that childbirth is a woman’s affair…”

Response #21: Thank you, done.

Comment #22: Page 11, line 185: “…accompanying their wife to a MWH to stay before delivery is a woman’s responsibility.” Your variable “accompanying wife to MWH for delivery is a woman’s responsibility” suggests that women deliver at the MWH, but is that true? Women are supposed to be moved to the health facility when labor comes so they deliver in the health facility. MWHs are used only for them to stay before delivery.

Response #22: Thank for the comment, we have revised it now.

Comment #23: Page 13, line 199-200: “56.3% of partners had decided to rest their spouse at a MWH…”

Response #23: Thank you, done.

Comment #24: Table 4: Variable “accompanied their spouse for resting in MWH for current child”. Use past tense since your other variables are past tense.

Response #24: Thank you, done.

Comment #25: Page 15, line 222: “MWH knowledge of male partners was positively associated…” 

Response #25: Thank you, done.

Comment #26: Page 15, line 221-222: “…associated with a 12% decrease…” AOR = 0.88 is written in text but in Table 5, AOR = 0.86 

Response #26: Thank you, done.

Comment #27: Page 15, line 225-226: “…associated with a 4.75 times increase…” AOR = 4.75 is written in text but in Table 5, AOR = 4.74 

Response #27: Thank you, done.

Comment #28: Page 16, line 242: “…assumed low because women are staying at a MWH until labor starts…” delete ‘s from “women’s”. 

Response #28: Thank you, done.

Comment #29: Page 17, line 271: “…whenever males are the primary decision makers in a family…” When it says “most decision makers” do you mean to say men are the primary decision makers in the household?

Response #29: Thank you, done.

Response to reviewer#3

Comment #1: The manuscript titled, “Factors affecting men’s involvement in maternity waiting home care in North Achefer district, Northwest Ethiopia: a cross-sectional study” requires additional copy editing for English grammar and language. Double check the correct terminology for use versus utilization throughout.

Response #1: Revised as per the comment.

Comment #2: Study variables and measurements: More details need to be provided for how the survey tool was developed. There are several questions in the supplementary file that are not addressed in the presentation of results. Variables are poorly defined. In section 2, wealth is extensively surveyed yet not described in the text. How were wealth quintiles defined? Were responses to questions in Sections 5,6,7 open ended or multiple choice? How were these survey questions derived? Was there any qualitative research guiding survey response choices? I’m not convinced that availing food when their spouse and relatives were at MWHs appropriately measures male partners’ involvement in MWHs. More details are needed about how Likert scales were determined and tested.

Response #2: The tool was developed by review of different literature. In the current revision, we have indicated in the introduction and in the method section, how male involvement has been defined and how we framed the male involvement indicators for the purposed of this study.

Regarding wealth index, now, we have clearly and briefly presented in the “Study variables and measurements” section how the wealth index was created. We have not done qualitative survey to set response options but we have developed through review of available literatures in the area. 

In fact, there is universally agreed definition regarding indicators of male involvement in any of reproductive, maternal and child health issues including maternity waiting home. In Ethiopia context, the MWHs are not equipped with basic facilities including kitchen facilities. In these cases, availing food can be a major challenge for pregnant women when they stay at MWH. Hence, we considered availing food as one indicator of male involvement.

Comment #3: Results: Justify the use of so many variables in one multivariable model. In Table 2, consider stratifying duration of stay at MWH and how many ANC visits a woman and spouse attended. Provide rationale for how attitudes were labeled positive or negative. Several results had very wide confidence intervals and potential reasons for the wide intervals need to be offered.

Response #3: We have used p-value less than 0.25 as cut value to select candidate variables for multivariable regression model. We assumed also the sample size is adequate to do so based on the assumption that 10-20 observations per variable considered in the multivariable regression.

We have reanalyzed the ANC and reclassified the length of stay as per recommendation. The possible reason for wider confidence interval may be due to small observation per cells for categorical variables. 

Comment #4: Discussion: Be sure to discuss limitations of the study. Elaborate on the cultural implications of men being the main drivers of decision making. Finally, propose next steps and tell the reader how interventions can “change patriarchal thinking in society” as suggested in the final sentence.

Response #4: we have made revision as per the comment.

---

## [Decision Letter · Decision Letter 1]

7 Dec 2021

PONE-D-20-07872R1Factors affecting men’s involvement in maternity waiting home care in North Achefer district, Northwest Ethiopia: a cross-sectional studyPLOS ONE

Dear Dr. Dabere Nigatu,

Thank you for submitting your manuscript to PLOS ONE. After careful consideration, we feel that it has merit but does not fully meet PLOS ONE’s publication criteria as it currently stands. Therefore, we invite you to submit a revised version of the manuscript that addresses the points raised during the review process.

Dear authors on your scholarly work; you have brought an important study problem in the area of practice. However, the manuscript has multiple language usage flaws including punctuations, wordings, spelling and mainly grammar errors. These problems are found throughout the manuscript. Moreover, there are several methodological limitations as the reviewers raised. Therefore, please make repeated proof-reading and thorough copyediting before resubmitting the manuscript. This would help increase the readability of the manuscript if published.

We look forward to receiving your revised manuscript.

Kind regards,

Wubet Alebachew Bayih, M.Sc.

Academic Editor

PLOS ONE

Additional Editor Comments (if provided):

Dear authors on your scholarly work; you have brought an important study problem in the area of practice. However, the manuscript has multiple language usage flaws including punctuations, wordings, spelling and mainly grammar errors. These problems are found throughout the manuscript. Moreover, there are several methodological limitations as the reviewers raised. Therefore, please make repeated proof-reading and thorough copyediting before resubmitting the manuscript. This would help increase the readability of the manuscript if published.

Reviewers' comments:

Reviewer's Responses to Questions

**Comments to the Author**

1. If the authors have adequately addressed your comments raised in a previous round of review and you feel that this manuscript is now acceptable for publication, you may indicate that here to bypass the “Comments to the Author” section, enter your conflict of interest statement in the “Confidential to Editor” section, and submit your "Accept" recommendation.

Reviewer #2: (No Response)

Reviewer #3: (No Response)

2. Is the manuscript technically sound, and do the data support the conclusions?

Reviewer #2: Yes

Reviewer #3: Yes

3. Has the statistical analysis been performed appropriately and rigorously? 

Reviewer #2: Yes

Reviewer #3: I Don't Know

4. Have the authors made all data underlying the findings in their manuscript fully available?

Reviewer #2: Yes

Reviewer #3: Yes

5. Is the manuscript presented in an intelligible fashion and written in standard English?

Reviewer #2: No

Reviewer #3: No

6. Review Comments to the Author

Reviewer #2: PONE-D-20-07872R1

“Factors affecting men’s involvement in maternity waiting home care in North Achefer district, Northwest Ethiopia: a cross-sectional study”

Reviewer Comments:

1.There are still a few grammatical errors. Some suggested changes are highlighted here:

a. Introduction, page 4, lines 76-77: “Thus, the current study applied an inclusive definition for male involvement in MVH, which includes male participation in deciding to rest their spouse to MWH…”

b.Methods, page 6, lines 122-123: “Then, the questionnaire was amended for wording, sequencing, and content…”

c.Methods, page 7, line 149: “The items used include male partner participation in deciding to rest their spouse to MWH…”

d.Results, page 12, line 208: “…should not require much attention from men…”

e.Discussion, page 18, line 285 and conclusion, page 18, line 301: “Men’s sole decision-making in family affairs…”

f.Discussion, page 18, line 289: “Men being primary drivers…”

g.Discussion, page 18, line 293: “The study has limitations.”

2.Results:

a.Typically, the sample size (n=403) is placed inside the table, for example under “frequency”, versus in the table title.

3.Discussion:

a.Page 17, line 270-273: When you say younger men/this age group, which age group are you referring to? In the results it says that a year increase in age was associated with a 14% decrease in the likelihood of men’s involvement – did you also do the multivariable logistic regression model by the age categories in Table 1?

b.Page 18, study limitations: Thank you for addressing my suggestion to discuss additional limitations of the study, specifically the generalizability of the study. However, when I say generalizability, I meant the degree to which your study findings are generalizable to other contexts outside North Achefer District or outside Ethiopia. So when you say “this study is generalizable to male partners whose spouse have used MWH…” is it only generalizable to North Achefer District, throughout Ethiopia, or can it be generalized to other similar contexts?

Reviewer #3: Comment #1: The manuscript titled, “Factors affecting men’s involvement in maternity waiting home care in North Achefer district, Northwest Ethiopia: a cross-sectional study” still requires additional copy editing for English grammar and language.

Comment #2: Update the abstract to include the definition of “male involvement” in the current study.

Also elaborate in the last sentence of the abstract to propose how to change “patriarchal thinking in society” to improve men’s involvement.

Authors state that the current study aimed to identify factors affecting men’s involvement in maternity waiting home care. Please define MWH care? Do the authors mean MWH use?

Comment #3: Study variables and measurements: More details need to be provided for how the survey tool was developed. There are several questions in the supplementary file that are not addressed in the presentation of results. Variables remain poorly defined. Were responses to questions in Sections 5,6,7 open ended or multiple choice? More details are needed about how Likert scales were determined and tested.

Comment #4: Results: Justify the use of so many variables in one multivariable model. Provide rationale for how attitudes were labeled positive or negative. Several results had very wide confidence intervals and potential reasons for the wide intervals need to be offered in the text not only as response to review comments.

7. PLOS authors have the option to publish the peer review history of their article (what does this mean?). If published, this will include your full peer review and any attached files.

Reviewer #2: No

Reviewer #3: No

---

## [Author Response · Author response to Decision Letter 1]

15 Jan 2022

Response to Academic Editor

Comment: Dear authors on your scholarly work; you have brought an important study problem in the area of practice. However, the manuscript has multiple language usage flaws including punctuations, wordings, spelling and mainly grammar errors. These problems are found throughout the manuscript. Moreover, there are several methodological limitations as the reviewers raised. Therefore, please make repeated proofreading and thorough copyediting before resubmitting the manuscript. This would help increase the readability of the manuscript if published.

Response: We tried to edit the entire document of the manuscript. We edited the spelling, space between words, and the whole grammar as much as possible with online grammar and language checkers (Quill bot online checker and Scribendi costumer service). The methodological concerns raised by the reviewers were adequately explained in the point-by-point response to reviewers.

Response to reviewers

Response to reviewer #2

Comment #1: There are still a few grammatical errors. Some suggested changes are highlighted here:

a. Introduction, page 4, lines 76-77: “Thus, the current study applied an inclusive definition for male involvement in MVH, which includes male participation in deciding to rest their spouse to MWH…”

b. Methods, page 6, lines 122-123: “Then, the questionnaire was amended for wording, sequencing, and content…”

c. Methods, page 7, line 149: “The items used include male partner participation in deciding to rest their spouse to MWH…”

d. Results, page 12, line 208: “…should not require much attention from men…”

e. Discussion, page 18, line 285 and conclusion, page 18, line 301: “Men’s sole decision-making in family affairs…”

f. Discussion, page 18, line 289: “Men being primary drivers…”

g. Discussion, page 18, line 293: “The study has limitations.”

Response #1: Thank you for suggesting editorial changes. We have revised the manuscript as per the comments. 

Comment #2: Results: a. Typically, the sample size (n=403) is placed inside the table, for example under “frequency”, versus in the table title.

Response #2: We have revised each table so as to include the sample size for each variable. 

Comment #3: Discussion: a. Page 17, line 270-273: When you say younger men/this age group, which age group are you referring to? In the results it says that a year increase in age was associated with a 14% decrease in the likelihood of men’s involvement – did you also do the multivariable logistic regression model by the age categories in Table 1?

Response #3a: Thank you for your deep insight about the variable age. Actually, we considered age as a continuous variable in the logistic regression model. Hence, the reviewer concern is very practical that we have revised the discussion in way it suits the result. 

b. Page 18, study limitations: Thank you for addressing my suggestion to discuss additional limitations of the study, specifically the generalizability of the study. However, when I say generalizability, I meant the degree to which your study findings are generalizable to other contexts outside North Achefer District or outside Ethiopia. So, when you say “this study is generalizable to male partners whose spouse have used MWH…” is it only generalizable to North Achefer District, throughout Ethiopia, or can it be generalized to other similar contexts?

Response #3b: Thank you for the comment. In fact, since the study follows scientifically sound methods, the findings are generatable to other settings in Ethiopia and outside Ethiopia with similar contexts. We have incorporated your concern in the updated version of the manuscript.

Response to reviewer#3

Comment #1: The manuscript titled, “Factors affecting men’s involvement in maternity waiting home care in North Achefer district, Northwest Ethiopia: a cross-sectional study” still requires additional copy editing for English grammar and language.

Response #1: We have revised the whole manuscript for grammar and language.

Comment #2: Update the abstract to include the definition of “male involvement” in the current study. 

Also elaborate in the last sentence of the abstract to propose how to change “patriarchal thinking in society” to improve men’s involvement.

Authors state that the current study aimed to identify factors affecting men’s involvement in maternity waiting home care. Please define MWH care? Do the authors mean MWH use?

Response #2: Thank you for the comment. The abstract is revised to include reviewer’s suggestions. Regarding MWH care Vs MWH use, the authors’ intention was to say MWH use. Thus, we have replaced the word “care” by “use” in the updated version of the manuscript. In the current revision, the tern “use” is replaced by “utilization” in the whole document for consistency and more reflectiveness. 

Comment #3: Study variables and measurements: More details need to be provided for how the survey tool was developed. There are several questions in the supplementary file that are not addressed in the presentation of results. Variables remain poorly defined. Were responses to questions in Sections 5,6,7 open ended or multiple choice? More details are needed about how Likert scales were determined and tested.

Response #2: Thank you for the comment. The survey tool was developed by reviewing different relevant literature. We have also done a pretest before the actual survey and we have amended the survey tool as per the feedback received during the pretest. In fact, there may be data assessed by the survey tool but not addressed in the results. However, for sure, we believe that we have addressed all the results that are relevant to the objective of the current study. 

The responses to questions in section 5, 6 and 7 were multiple choice. In some cases, more than one answer is possible and questions with such response options were clearly indicated in parenthesis. 

Upon literature review, there are studies employing different point Likert scales: two-point scale, three-point scale, four-point scale, five-point scale, seven-point scale and even more than seven-point Likert scale (Chang, 1997, Vagias, 2006, Taherdoost, 2019). The use of below five-point scale could be very conservative or provide restrictive options for raters to choose while the use of seven or more than seven-point scale make rating complex and get sophisticate raters and usually recommended for well-educated raters. Literature also suggests that five-point scale appears to be less confusing and to increase response rate. It is also quite simple for interviewer to read out the complete list of scale descriptors as compared to seven-point or more than seven point Likert scales (Bouranta et al., 2009, Dawes, 2008). Thus, we preferred to use five-point scale since it provides freedom for raters and because of simplicity for rating. Additionally, the current study is a community-based study. In community-based studies where you can find people with diverse level of literacy, the use of a 7-point scale or more than a 7-point scale is not recommended. At the end, the pretest has helped us to check and decide about the understandability and simplicity a five-point Likert scale for raters and interviewers. 

Comment #4: Results: Justify the use of so many variables in one multivariable model. Provide rationale for how attitudes were labeled positive or negative. Several results had very wide confidence intervals and potential reasons for the wide intervals need to be offered in the text not only as response to review comments.

The number of variables used in multivariable model were 14. As inclusion criteria, we have used p-value less than 0.25 as cut value to select candidate variables for multivariable regression model. The sample size is also adequate to do so based on the assumption that 10-20 observations per variable considered in the multivariable regression. For example, if we calculate required minimum sample size by taking the upper bound of the assumption (i.e., 20 observations per variable), it will give us 280 (14*20), which is less than the actual sample size. 

Regarding the rationale for labeling positive or negative attitude, as a matter of fact, we can find studies that label attitude as favorable or unfavorable and at the same time, we can find studies that label attitude as positive or negative by using attitude mean score or median score. In this study, eight items with a five-point Likert scale were used to assess men’s attitudes towards MWH. Hence, those men who scored above the median were considered to have positive attitude towards MWH use otherwise taken as having negative attitude (Fang et al., 2021, Seid and Hussen, 2018, Dahake and Shinde, 2020). The rational for labeling attitude positive or negative is just to show inclination towards certain behavior. In the case of this study, positive attitude towards MWH utilization means they have inclination to use MWH while negative attitude reflects the behavioral inclination not to use MWH. 

Potential reasons for wide confidence interval may include sample size, multicollinearity, and presence of cells with observation very closer to zero. In this study, multicollinearity was checked by variance inflation factor and had no role for the wide confidence intervals observed in the study. Still, the wide confidence intervals might be associated with sample size adequacy and cell with small observation. For example, the variable decision-making in family affair had one cell with small observation. Having said these all, the authors believe that the confidence intervals recorded in this study are not worse to affect the conclusions. 

Reference 

BOURANTA, N., CHITIRIS, L. & PARAVANTIS, J. 2009. The relationship between internal and external service quality. International Journal of Contemporary Hospitality Management, 21, 275-293.

CHANG, L. 1997. Dependability of anchoring labels of Likert-type scales. Educational and Psychological Measurement, 57, 800-807.

DAHAKE, S. & SHINDE, R. 2020. Exploring Husband's Attitude Towards Involvement in his Wife's Antenatal Care in Urban Slum Community of Mumbai. Indian journal of community medicine : official publication of Indian Association of Preventive & Social Medicine, 45, 320-322.

DAWES, J. 2008. Do data characteristics change according to the number of scale points used? An experiment using 5-point, 7-point and 10-point scales. International journal of market research, 50, 61-104.

FANG, Y., LIU, P. & GAO, Q. 2021. Assessment of Knowledge, Attitude, and Practice Toward COVID-19 in China: An Online Cross-Sectional Survey. The American journal of tropical medicine and hygiene, 104, 1461-1471.

SEID, M. A. & HUSSEN, M. S. 2018. Knowledge and attitude towards antimicrobial resistance among final year undergraduate paramedical students at University of Gondar, Ethiopia. BMC Infectious Diseases, 18, 312.

TAHERDOOST, H. 2019. What Is the Best Response Scale for Survey and Questionnaire Design; Review of Different Lengths of Rating Scale / Attitude Scale / Likert Scale. International Journal of Academic Research in Management (IJARM), 8.

VAGIAS, W. M. 2006. Likert-type scale response anchors. Clemson International Institute for Tourism & Research Development, Department of Parks, Recreation and Tourism Management. Clemson University.

---

## [Editor Report · Decision Letter 2]

28 Jan 2022

Factors affecting men’s involvement in maternity waiting home utilization in North Achefer district, Northwest Ethiopia: a cross-sectional study

PONE-D-20-07872R2

Dear Dr. Dabere Nigatu,

We’re pleased to inform you that your manuscript has been judged scientifically suitable for publication and will be formally accepted for publication once it meets all outstanding technical requirements.

Kind regards,

Wubet Alebachew Bayih, M.Sc.

Academic Editor

PLOS ONE
---

## [Editor Report · Acceptance letter]

2 Feb 2022

PONE-D-20-07872R2 

Factors affecting men’s involvement in maternity waiting home utilization in North Achefer district, Northwest Ethiopia: a cross-sectional study 

Dear Dr. Nigatu:

I'm pleased to inform you that your manuscript has been deemed suitable for publication in PLOS ONE. Congratulations! Your manuscript is now with our production department. 

Kind regards, 

on behalf of

Dr. Wubet Alebachew Bayih 

Academic Editor

PLOS ONE